# N-Student Learning: An Approach to Model Uncertainty and Combat Overfitting

## Abstract

This work presents N-Student Learning, a pseudo-label based multi-network training setup that can be applied to nearly any supervised learning architecture in order to help combat the problem of overfitting and control the way in which a network models uncertainty in the data. The effectiveness of N-Student Learning relies on the idea that a network's predictions on unseen data are largely independent of any instance-dependent noise in the labels. In N-Student Learning, each student network is assigned a subset of the training dataset such that no data point is in every student's training subset. Unbiased pseudo-labels can thus be generated for every data point in the training set by taking the predictions of appropriate student networks. Training on these unbiased pseudo-labels minimizes the extent to which each network overfits to instance-dependent noise in the data. Furthermore, based on prior knowledge of the domain, we can control how the networks learn to model uncertainty that is present in the dataset by adjusting the way that pseudo-labels are generated. While this method is largely inspired by the general problem of overfitting, a natural application is found in the problem of classification with noisy labels — a domain where overfitting is a significant concern. After developing intuition through a toy classification task, we proceed to demonstrate that N-Student Learning performs favorably on benchmark datasets when compared to state-of-the-art methods in the problem of classification with noisy labels.

## 1 Introduction

Overfitting is a fundamental problem in supervised classification in which a model learns properties of the training data that do not generalize to unseen data. If samples from the input space contain information that is not relevant to the task, the model may overfit by learning to use these irrelevant details for predictive purposes. Overfitting may also occur as a result of noise in the label space, in which the label provided in the dataset does not match the expected output of the model given the corresponding input. Overfitting due to label noise in its various forms will be a primary focus of this paper.

In this paper, we introduce N-Student Learning, a pseudo-label based multi-network training setup which mitigates overfitting to noise in the labels. The idea is to relabel the dataset by taking the predictions of networks that have never seen the data. We do this by training multiple networks on different subsets of the data so that the pseudo-labels that they generate on their respective unseen subsets will be clean of any instance-dependent noise. Training on these pseudo-labels results in networks that are less prone to overfitting.

After introducing the architecture in section 2, we will discuss a few types of label noise that are commonly present in datasets. Using a toy classification problem in section 3, we show the effect of the N-Student Learning setup and demonstrate that the setup can be adapted to handle different kinds of noise. Following this, in section 4, we show that N-Student Learning performs favorably when compared to state-of-the-art methods on both artificially noisy and naturally noisy benchmark datasets.

## 2 METHOD

### 2.1 TRAINING SETUP

Supervised classification is the problem of learning a function that maps an input space $X$ to a label space $Y$, given a dataset $D = \{x_i, y_i\}_{i=1}^{S}$, where $x_i \in X$ and $y_i \in Y$, and is the $S$ is the size of the dataset.

In N-Student Learning, we train $n$ student networks $\{N_i\}_{i=1}^{n}$ in parallel, where each network $N_i$ trains on a subset $D_i$ of the dataset. In our work, it is assumed that these networks all have the same architecture. These subsets are generated with the constraint that for each sample $(x_i, y_i)$ in the dataset, there is some network that does not train on that sample. This allows us to generate clean pseudo-labels for each sample. Thus, the following must hold:

$$D = \bigcup_{i \in \{1,2,\ldots,n\}} D_i^c, \text{ where } c \text{ denotes a set complement.}$$

We begin training with a warmup phase where we train each $N_i$ on its respective dataset $D_i$ for a small number of epochs without pseudo-labels. After the warmup phase, each student continues to train on its own subset $D_i$, but with a portion of the labels replaced by pseudo-labels generated by a student that has never seen the respective inputs. We define a hyperparameter $p$, the pseudo-label rate, which controls what proportion of the ground-truth labels will be replaced by pseudo-labels. At the beginning of each epoch, we randomly select a subset $D_{\text{sample}}$ of size $\lfloor pS \rfloor$. For each pair $\{x_i, y_i\} \in D_{\text{sample}}$, we replace it with $\{x_i, N_j^*(x_i)\}$, where $N_j$ is a network that does not train on $x_i$ and $N_j^*(x_i)$ is a pseudo-label generated from the prediction of $N_j$ on $x_i$. After this pseudo-labeled dataset is generated, we train each student on its respective pseudo-labeled subset.

We experiment with three methods for generating pseudo-labels: hard, soft, and stochastic pseudo-labels. Hard pseudo-labels are generated by taking the argmax of the predicted logits and assigning that class as the correct label in the form of a one-hot vector. Soft pseudo-labels are generated by using the distribution defined by the output of the softmax layer as the target vector. Lastly, we define stochastic pseudo-labels to be pseudo-labels generated by sampling from the distribution output by the network and representing the sampled class as a one-hot vector.

We restrict our experiments to cases in which each student is allocated a training subset of equal size. For 2 students, this means that each student trains on half of the data, and therefore generates pseudo-labels for the other half. For 3 students, each student can train on anywhere from one third to two thirds of the data with this equal-allocation restriction. To further constrain the choice of subsets, we only consider splits in which the fraction of the dataset per student is on the boundary of the possible range of values. For $n$ students, this means that each student trains on either $\frac{1}{n}$ or $\frac{n-1}{n}$ of the data. This is because if each student trains on less than $\frac{1}{n}$ of the dataset, then part of the dataset is not used for training because $|\bigcup_{i \in \{1,2,\ldots,n\}} D_i| \leq n|D_0| < |D|$. If each student trains on more than $\frac{n-1}{n}$, then each student can pseudo-label less than $\frac{1}{n}$ of the dataset, which is not enough to label the whole dataset.

In the case that each student trains on $\frac{1}{n}$ of the data, the training subsets are disjoint and for any student's subset, each of the $n - 1$ other networks are able to pseudo-label the samples from that subset. Each epoch, we choose a random network to pseudo-label these samples. If each student trains on $\frac{n-1}{n}$ of the data, then there is only one network that can pseudo-label each sample, so there are no choices to be made.

There are a variety of ways to perform inference when training multiple networks. One way is to choose a random network after training to use for inference. Alternatively, predictions can be aggregated by averaging or multiplying class-wise before normalizing, which can improve performance.

---

Important hyperparameters:
    $n$: Number of students
    $p$: Pseudo-label rate (proportion of labels that are replaced by pseudo-labels each epoch)

## 3 REPRESENTING UNCERTAINTY

Although the goal of supervised classification is to predict correct labels, we often want a classifier whose predictions reflect the correct conditional probability distribution over the labels given an input sample—not just the highest likelihood class. In the following paragraphs, we will describe a few types of classification problems and how they may impact the training of an effective classifier.

Consider a problem in which a label can determined with near certainty for any given input sample. An example of such a domain may be the classification of unambiguous high-resolution images. In this case, learning the distribution of the dataset and fitting to the dataset's particular samples may be sufficient to effectively model the problem.

An interesting case arises when working with domains which contain inherent stochasticity. An example of such a case is a future prediction problem, such as the problem of predicting whether a house will be sold using only information about features of the house. Given information about any particular house, you can never have absolute certainty that the house will be sold or not. Despite this uncertainty, the dataset that we are provided contains binary labels representing whether a particular house has been sold. The main distinction between this case and the last is that in the stochastic problem, each instance of data in the training set may not be representative of the correct distribution that you want to learn. However, similar to the previous unambiguous case, the dataset as a whole may be representative of the correct distribution of the problem.

Finally, consider the case of an unambiguous problem that simply contains errors in the labels. In this case, both the individual samples and the dataset as a whole may not be fully representative of the task. We explicitly do not want to model the exact distribution that is found in the data. Instead, we can ignore these errors by modeling the most likely class label, filtering out any uncertainty. This is also precisely the problem of training with noisy labels.

A key insight is that without prior knowledge about the domain and the way that the dataset was generated, these three cases may be completely indistinguishable. Despite this, we may want to represent uncertainty that is present in these datasets in different ways depending on the type of noise that is present. In section 3.1, we will show that based on the priors that one has on a dataset, N-Student Learning can be used to appropriately model the uncertainty in the data by adjusting the way that pseudo-labels are generated.

### 3.1 A TOY PROBLEM: CLASSIFICATION OF TWO GAUSSIANS

In this section, we demonstrate the impact of N-Student Learning on overfitting through a toy classification problem. We also show how the choice of pseudo-labeling method impacts the network's outputs. The approaches will be compared by visualizing network predictions across the 2-d input space.

#### 3.1.1 TWO GAUSSIANS: EXPERIMENTAL DESIGN

The dataset for this classification problem is generated by sampling 1000 total points from two circular Gaussians separated by 2 units, each with unit variance. In fig. 1, we see the data sample.

The networks that we use for the baseline and for the 2-Student setups are standard feed forward networks with 5 hidden layers of 100 neurons and ReLU activation. They are trained for 400 epochs with stochastic gradient descent with 0.9 momentum and batch size 16.

In order to demonstrate the effect of each training setup, we visualize the output of the network by displaying the predicted class probabilities for each point on the input space. These probabilities are represented using a gradient between blue and yellow in fig. 2 and fig. 3.

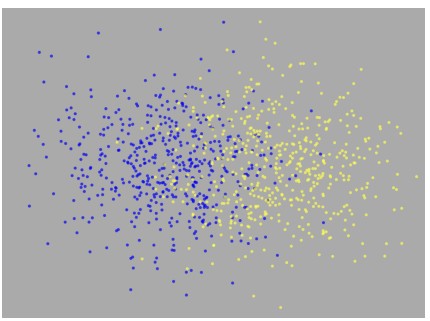

Figure 1: A dataset of 1000 total points generated by sampling from two circular Gaussians.

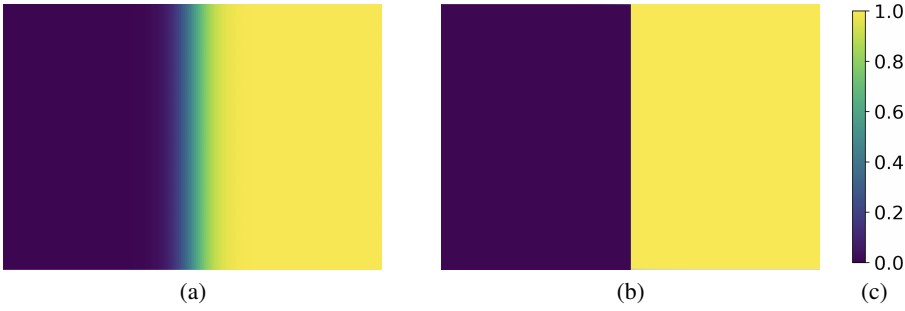

Figure 2: (a) shows the conditional probability that a point is drawn from each Gaussian. (b) shows the most likely class given any data point. (c) shows the mapping between color and class probability.

### 3.1.2 Two Gaussians: Results

We see in fig. 3 that the baseline approach overfits to the training set by epoch 400. In contrast, the 2-Student setups are more robust to overfitting.

Soft and stochastic pseudo-labels generate predictions that are fairly representative of the actual conditional probability that a point belongs to each Gaussian which can be found in fig. 2 (a). Hard pseudo-labels form sharp decision boundary down the middle, more akin to fig. 2 (b).

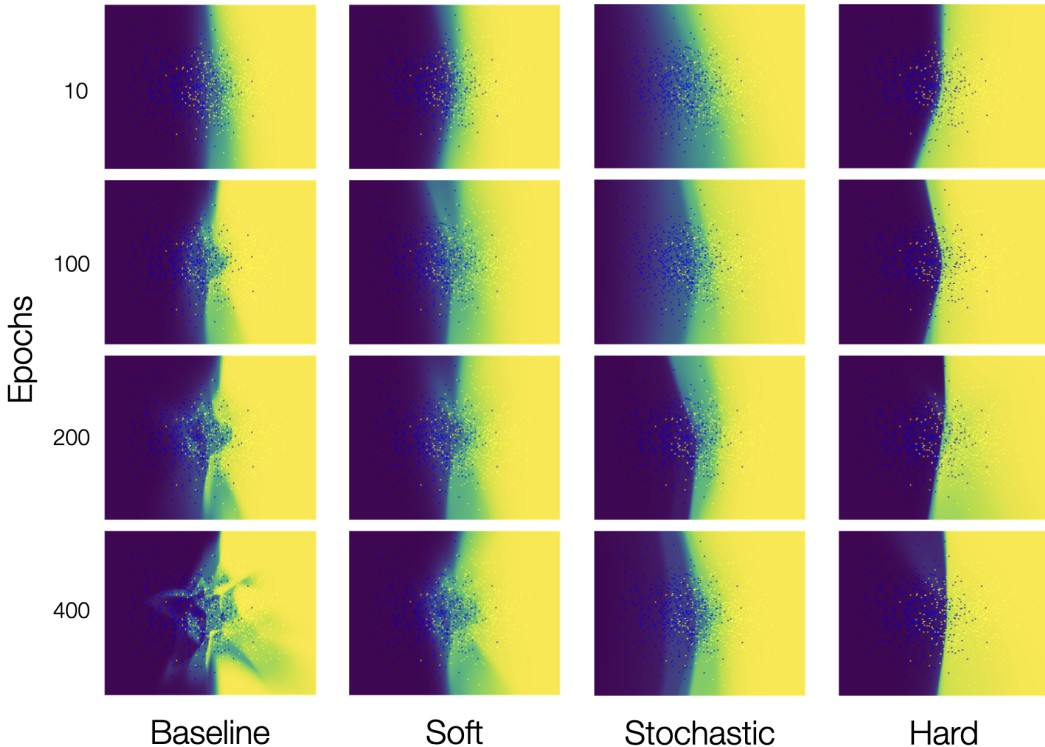

Figure 3: The outputs of a baseline network are compared to those of 2-Student setups trained using soft, stochastic, and hard pseudo-labels. The outputs are visualized at epochs 10, 100, 200, and 400. The color map is the same as fig. 2.

### 3.1.3 Two Gaussians: Discussion

Without a prior on how the dataset is generated, you cannot distinguish between different explanations for the uncertainty that appears on the boundary between the two Gaussians. In this case, the true generation process is fundamentally stochastic and the uncertainty in the middle results from a stochastic sampling process. However, it is also plausible that the two classes modeled in the dataset are completely disjoint in nature, separated by a clear-cut decision boundary. In this case, the uncertainty may be explained by errors in label generation that are more likely to occur towards the boundary. The optimal network outputs for these two cases are shown in fig. 2.

A useful feature of N-Student Learning is that one can use their prior on the type of uncertainty that is present in a dataset to choose a pseudo-labeling method that appropriately represents the uncertainty. If the problem is known to be fundamentally stochastic and minimally noisy, then we want to use soft or stochastic pseudo-labels in order to effectively model the conditional distribution over the classes given a data point. On the other hand, if we have reason to think that the dataset contains a lot of errors in the labels, perhaps as a result of the labeling process, then we can effectively filter out the errors near the decision boundary by using hard pseudo-labels.

## 4 Noisy Labels

The task of classifying with noisy labels has been widely researched and has many applications. In addition to being a great task to study the problem of overfitting, learning to train with noisy labels also has the potential to reduce our reliance on clean, human-labeled datasets. There is a trade-off between cost and accuracy when creating the datasets that we train on. At one end, we have expensive carefully human-labeled datasets and on the other, we have inexpensive automatically generated datasets full of noise. Properly leveraging the latter efficient and scalable datasets requires training techniques that are more robust to noise in the labels.

### 4.1 Related Works

#### 4.1.1 Sample Selection Methods

A popular class of methods for training on noisy data involves sample selection, a technique in which clean instances of data are selected by relying on the "low-loss criterion", the principal that when a trained network predicts on the training set, lower loss samples are more likely to have clean labels. Below, we discuss a number of sample selection related approaches.

**MentorNet** is an approach that connects the problem of noisy data to curriculum learning (Jiang et al., 2018), a sub-field that looks to generate a curriculum for networks to train on. MentorNet first trains a teacher network on noisy data. This teacher network is then used to generate a curriculum for a student network to learn off of—a dynamic weighting of the importance of the samples in the dataset.

**Decoupling** trains two networks in parallel with an "Update by Disagreement" algorithm in which the networks are trained on the samples that the two networks disagree on (Malach & Shalev-Shwartz, 2017).

**Co-teaching** similarly adopts a two-network setup (Han et al., 2018). Each epoch, the low-loss samples from each network are chosen to train the other network.

**Co-teaching+** combines the Update by Disagreement algorithm from Decoupling and the low-loss criterion (Yu et al., 2019). Accordingly, the networks train on samples where the networks disagree and have a small loss.

**JoCoR** trains two networks in parallel on low-loss samples where the predictions of the two networks align (Wei et al., 2020). The idea is that a sample is more likely to be clean if the networks agree in addition to the samples having a low loss.

### 4.1.2 SEMI-SUPERVISED AND SELF-SUPERVISED METHODS

A class of methods reframes the noisy label problem as a semi-supervised learning problem in which low-loss samples are used to split the dataset into a clean and unlabeled dataset in order to leverage existing semi-supervised learning methods. Self-supervised methods are also particularly well-suited for the noisy-label problem because self-supervised pre-training is not impacted by noise in the labels.

**DivideMix** is a semi-supervised method which trains two networks(Li et al., 2020). Every epoch, semi-supervised methods are used on the labeled-unlabeled split generated by the other network. The semi-supervised methods used by DivideMix include techniques such as generating pseudo-labels by averaging and sharpening the predictions of a network over multiple augmentations. The mixup algorithm is also used, which encourages linear behavior in a network by training on synthesized interpolated training examples that are generated by taking a linear combination of both the input images and the labels.

**Co-learning** is a self-supervised approach that takes further advantage of multiple augmentations for training (Tan et al., 2021). A weakly augmented view and two strongly augmented views of the input are generated and passed through a shared encoder, where the encoded weakly augmented view is passed through a classification head for a traditional classification loss. The two strongly augmented views are passed through a projection head to a space where a contrastive loss is applied, creating a stronger encoder for the classification head to use. Finally, a loss is added that encourages the property that for any two given samples, the classifier head and the projection head's encodings have a similar distance. All of these properties work together to create a robust classifier.

### 4.2 EVALUATION

Noisy label methods are commonly evaluated on both artificially corrupted datasets and datasets with natural noise. We will be evaluating on three artificially corrupted datasets: MNIST, CIFAR-10, and CIFAR-100, as well as one naturally corrupted dataset, ANIMAL-10n (Song et al., 2019).

### 4.2.1 ARTIFICIALLY NOISY DATASETS

Artificially corrupted datasets are supervised classification benchmark datasets that have noise added to them. Two common types of noise injection methods are symmetric noise and asymmetric noise.

Symmetric noise is generated by randomly corrupting each label into a different label with some probability. As an example, symmetric noise with a flip rate of 0.4 means that every data point's label will be flipped to a different label with 40% probability.

Asymmetric noise is designed to mimic class-dependent noise, in which one class may get confused with another. This can be implemented in a number of ways. For MNIST and CIFAR-100, past works implement asymmetric noise by corrupting each class into the next class with some probability. Following previous works, "next" is arbitrarily chosen to be the class represented by the following index in the target class vector.

For asymmetric noise on CIFAR-10, past works corrupt the classes in a more meaningful way to mimic real noise (Arpit et al., 2017). Specifically, we have the following mappings: { bird $\rightarrow$ airplane, cat $\rightarrow$ dog, deer $\rightarrow$ horse, truck $\rightarrow$ automobile } Note that only 4 of the 10 classes are corrupted here, so in the case of CIFAR-10, the effective noise is two fifths of what is typically expected for a particular noise level.

### 4.2.2 NATURALLY NOISY DATASETS

While artificially corrupted datasets are a good domain for experimentation, we would also like to perform well on datasets that naturally have noise, as these will be the most reflective of real-world datasets.

We report results on ANIMAL-10n, a dataset that contains 5 pairs of confusing animals that are human labeled with an estimated noise level of 8%.

## 4.3 COMPARISONS

### 4.3.1 CIFAR-10, CIFAR-100

On popular benchmarks such as CIFAR-10 and CIFAR-100, we do not compare to methods for which the baseline performance (without added noise) is significantly stronger than a baseline network trained with a cross-entropy loss. These are typically methods that make use of strong augmentations and contrastive learning.

The approaches that we compare to are the following: Baseline, Decoupling, Co-teaching, Co-teaching+, and JoCoR.

Although we do not compare to Co-learning due to its use of contrastive learning, we choose to include Co-learning in table 2 and table 3 as a reference due to the relative low computational complexity compared to other methods that we omit.

Following previous multi-network approaches, our results take the predictions of a single student in a 2-Student architecture.

### 4.3.2 ANIMAL-10N

We compare to all public results on ANIMAL-10n. These include the following: Baseline, Co-Learning, Nested Co-Teaching (Chen et al., 2021), PLC (Zhang et al., 2021), SELFIE (Song et al., 2019), and S3 (Feng et al., 2022)

Many of these methods are compute heavy, and so we train a 4-Student setup with multiplicative-aggregate predictions.

## 4.4 EXPERIMENTAL SETUP

We use the same architectures as previous works. For CIFAR-10 and CIFAR-100, we follow Co-learning. For ANIMAL-10n, we follow SELFIE. Details can be found in table 1.

Table 1: Training Setups

| Dataset | CIFAR-10, CIFAR-100 | ANIMAL-10n |
|---|---|---|
| Architecture | ResNet-18 | VGG-19 |
| Warm-up Epochs | 10 | 10 |
| Epochs | 200 | 100 |
| Optimizer | Adam | SGD, momentum = .9 |
| Learning Rate | .001, linear decay from epoch 80 | .01, linear decay from epoch 40 |
| Augmentations | Weak (Rotation, translation) | Weak (Rotation, translation) |

Table 2: CIFAR-10 Results (% Accuracy)

| Methods | Noise Type/Rate | | | |
|---|---|---|---|---|
| | Symmetric 20% | Symmetric 50% | Symmetric 80% | Asymmetric 40% |
| Baseline | 84.81 | 61.49 | 28.98 | 76.30 |
| Decoupling | 85.75 | 61.93 | 27.23 | 74.97 |
| Co-teaching | 90.29 | 63.45 | 28.03 | 74.25 |
| Co-teaching+ | 88.63 | 76.27 | **30.37** | 81.25 |
| JoCoR | **90.43** | 66.00 | 29.19 | 73.95 |
| *Co-learning | 92.21 | **84.49** | **61.20** | 81.42 |
| 2-Students | 86.22 | **81.29** | 24.77 | **88.59** |

Table 3: CIFAR-100 Results (% Accuracy)

| Methods | Noise Type/Rate | | | |
|---|---|---|---|---|
| | Symmetric 20% | Symmetric 50% | Symmetric 80% | Asymmetric 40% |
| Baseline | 57.79 | 33.75 | 8.64 | 42.49 |
| Decoupling | 56.18 | 31.58 | 7.71 | 41.51 |
| Co-teaching | **64.28** | 32.62 | 6.65 | 40.62 |
| Co-teaching+ | 55.40 | 26.49 | 8.57 | 38.98 |
| JoCoR | 62.29 | 30.19 | 6.84 | 39.72 |
| *Co-learning | **66.58** | **54.54** | **35.45** | 41.96 |
| 2-Students | 53.58 | **40.58** | **15.30** | **48.65** |

## 4.5 RESULTS

### 4.5.1 CIFAR-10

In table 2, we see that on the 2-Student setup performs comparably to other methods on 20% symmetric noise, outperforms other methods on 50% symmetric noise, but underperforms on 80% symmetric noise. Notably, the 2-Student setup outperforms all methods by a large margin on asymmetric noise with 40% noise, including Co-learning.

### 4.5.2 CIFAR-100

In table 3, we see that our method outperforms all other methods in every metric other than 20% symmetric noise, where we underperform. Again, we find that the 2-Student setup is particularly strong in the domain of asymmetric noise, where we outperform all other methods.

### 4.5.3 ANIMAL-10N

In table 4, we see that our method outperforms all methods other than S3, a method with remarkably strong baselines due to self-supervised learning.

We also see that the performance of N-Student Learning improves as we increase the number of students, as demonstrated in section 4.5.4.

Table 4: ANIMAL-10n Results (% Accuracy)

| Methods | Accuracy |
|---|---|
| Baseline | 79.40 |
| 2-Student Setup | 81.22 |
| SELFIE | 81.80 |
| Co-Learning | 82.18 |
| 3-Student Setup | 83.70 |
| PLC | 83.40 |
| Nested Co-Teaching | 84.10 |
| 4-Student Setup | 84.57 |
| S3** | **88.50** |

---

* indicates a method that we do not compare to as discussed in section 4.3.1
** indicates a pre-print at the time of writing

Table 5: Ablation Over Hyperparameters on CIFAR-10 (% Accuracy)

| $n$ | Subset Proportion | Single Student | Aggregate (Multiplicative) |
|---|---|---|---|
| 2 | 1 / 2 | 81.29 | 82.27 |
| 3 | 1 / 3 | 75.66 | 82.74 |
| 3 | 2 / 3 | 82.04 | 83.98 |
| 4 | 1 / 4 | 72.38 | 82.25 |
| 4 | 3 / 4 | 82.20 | 84.72 |

#### 4.5.4 VARYING N

In this experiment, we show how the performance of N-Student Learning is affected by the number of students and the amount of data that each student trains on, and the inference method. We test on CIFAR-10 with 50% noise and $p = 0.5$. We vary $n$ from 2 to 4 and test the upper and lower bounds of the range of subset sizes that a student can train on: $\frac{1}{n}, \frac{n-1}{n}$.

The results can be found in table 5. Unsurprisingly, we find that aggregating the predictions of students is stronger than any isolated student's performance. For n = 3 and n = 4, we find that training each student on a larger subset performs better; the best performance achieved is with 4 students, each training on $\frac{3}{4}$ of the data, reaching a validation accuracy of 84.72%, which is higher than Co-learning.

### 4.6 DISCUSSION

In our results, we find that N-Student Learning performs particularly well on the ANIMAL-10n dataset as well as on the asymmetric noise tasks. The strong performance on asymmetric noise is important because real noise commonly contains similar class-dependant asymmetries.

The area that N-Student Learning consistently underperforms in is 20% symmetric noise. We hypothesize that this is due to the lower baseline performance of the 2-Student setup due to the data limitation per student. This may be less of a concern when training on large real-world datasets, where even half of the dataset contains enough samples to effectively learn the task.

We find that this simple multi-network architecture outperforms state-of-the-art approaches on benchmark noisy datasets in settings that are most similar to natural noise. Our results demonstrate that filtering samples by a low-loss criterion is not necessary for good performance on noisy labels.

## 5 CONCLUSION AND FUTURE WORKS

N-Student Learning is a flexible training setup that allows for control over the way that a network learns to model uncertainty that is present in the data. It is a meta-architecture that can be applied on top of essentially any supervised learning architecture to allow for robust training in the face of overfitting.

The training setup of N-Student Learning can be modified in various ways that may yield superior performance. Notably, the architecture and the size of the training subsets for each student are held constant for each experiment. However, it seems likely that other setups may be effective. As an example, consider training a strong student with a more powerful architecture and a larger subset of the data, supported by a set of weaker students cleaning the labels for the stronger student.

N-Student Learning can potentially be applied to tasks such as semi-supervised learning. In semi-supervised learning, in addition to splitting the labeled training set, the unlabeled portion of the dataset could be split as well. This could help minimize self-confirmation bias and prevent overfitting to the limited number of labeled samples.

It is our hope that the discussion on label uncertainty and the ideas behind N-Student Learning will inspire further work on a variety of problems.

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
