# OpenReview forum: "N-Student Learning: An Approach to Model Uncertainty and Combat Overfitting"
_ICLR.cc/2023/Conference — Submitted to ICLR 2023_

### Official Review · Reviewer_X44G · 2022-10-23

**Confidence:** 4
**Correctness:** 3
**Technical Novelty And Significance:** 1
**Empirical Novelty And Significance:** 2
**Recommendation:** 3

**Clarity, Quality, Novelty And Reproducibility:**

While the proposed method and presentation is clear overall, I believe this work lacks rigor and depth. It seems to be a simple adaptation of label propagation for ensemble learning, and there is very little analysis about *why* this method should work, and *how* it is advantageous over existing methods.

**Strength And Weaknesses:**

Strength:
- Writing is clear overall and readability is good.
- Empirical results show promise of method.

Weakness:
- Analysis and explanation of method is lacking.
    - The empirical results are promising, but there is little to no intuition for why this method works.
    - E.g. while the authors stress the importance of pseudo-labelling only for datapoints a network has never seen, and that this produces "clean" labels which are "clean of any instance-dependent noise", there is little to no discussion/reasoning/analysis on why this is so significant. At the very least, there should be a discussion/ablations on the  of the pitfalls of pseudo-labelling for previously seen datapoints.
    - Further, the proposed method seems to depend on each network successfully learning general representations of the data, such that the pseudo-labels reflect the underlying structure in the data. What mechanisms do you have that will enforce this will occur when each network is still learning with noisy labels? Or is the performance of the proposed method simply an empirical finding?
- Details and rigor lacking overall.
    - In all of the experiments, what is the "baseline" method?
    - Are all of the results reported for just one seed? No repeated trials and standard errors?
    - The authors mention 3 ways of generating pseudo-labels: hard, soft, stochastic. Which labeling method was used for the evaluations in Section 4.2?
    - What is the exact formula to multiply predictions to aggregate them during inference? Do you multiply class probabilities, then normalize the output again into a probability? If so, what is the rationale for such an equation?

**Summary Of The Paper:**

This paper proposes an ensemble based label-propagation method for classification in the presence of label noise. In the method, the authors emphasize the importance of generating pseudo-labels only for datapoints a model has not seen during training. Empirical comparisons against existing algorithms are limited but promising.

**Summary Of The Review:**

The proposed method is quite simple and could be promising, but I believe this paper in its current format is not fully convincing and lacks rigor and scientific significance. I find this to be mostly an empirical work that seems also rather light on the empirical findings and not extensive on the range of experiments and ablations performed. I would like to encourage the authors to build on this with either more analysis or empirical studies.

---

### Official Review · Reviewer_Riap · 2022-10-24

**Confidence:** 4
**Correctness:** 3
**Technical Novelty And Significance:** 2
**Empirical Novelty And Significance:** 2
**Recommendation:** 3

**Clarity, Quality, Novelty And Reproducibility:**

The paper is clear and the proposed method is well described. The experiments conducted seem to contain sufficient details to reproduce.

**Strength And Weaknesses:**

Strength:
- Overall, the paper is well-written and easy to follow.
- The proposed method is technically sound.
- The experiments conducted are reasonable, and the proposed method can achieve good performance.

Weaknesses:
- My main concern regarding the paper is the lack of novelty. The use of pseudo-labels has been previously demonstrated to be effective to combat the issue of noisy labels. The proposed method of using multiple models each seeing only a subset of the entire training set also seems to be a rather straightforward idea. The main contribution of the paper is arguably the experimental comparison against several previously proposed methods for learning with noisy labels. I am not sure if this would be technically novel enough to be published. It might be more interesting if the authors of the paper can also offer some theoretical insights on the effect of multiple networks each one only seeing a subset of the training dataset.
- Another concern I have is the lack of an ablation study. For instance, what if we had the same number of models, but each trained using the entire dataset? Would that be worse than the proposed method?

**Summary Of The Paper:**

This paper proposes an interesting "N-student learning" framework that can be used to tackle the framework of learning with noisy labels. Specifically, the method involves training multiple neural networks, each seeing only a non-overlapping subset of the entire training dataset. The authors of the paper also take one step further by adopting the use of pseudo-labels to further make the training more effective. They demonstrate that this simple method can achieve very good performance on several benchmark datasets tested.

**Summary Of The Review:**

All in all, due to the weaknesses mentioned above, I recommend rejecting the paper for now.

---

### Official Review · Reviewer_JspX · 2022-10-24

**Confidence:** 4
**Correctness:** 2
**Technical Novelty And Significance:** 2
**Empirical Novelty And Significance:** 2
**Recommendation:** 3

**Clarity, Quality, Novelty And Reproducibility:**

The quality and novelty of the paper are noted above. The clarity of the paper may also need to be improved. For example, the beginning of Section 3 mainly talks about the background of uncertainty and appears irrelevant to me. It would be better to move it to the related work and leave a brief introduction there.

Also, I believe the entire section 3 wishes to demonstrate the capability of the proposed method in learning uncertainty. However, except for some visual presentations of the decision boundaries on a toy dataset, there is no quantification of the uncertainty learning performance. Therefore, it is hard to judge whether the proposed method can truly learn uncertainty better.

**Strength And Weaknesses:**

Strength:
* The paper is overall organized well and easy to follow.



Weakness:
* The paper is lacking novelty. Leveraging the cooperation between multiple networks to combat overfitting and label noise is not new. There exists a handful of works following this vein such as Co-teaching, DivideMix, and Co-learning, as also mentioned by the authors in the related work section. However, there is no detailed discussion of the differences between the proposed method and existing methods.

* The quality of the work needs to be improved. Since the proposed method is claimed to be generally applicable and is similar to existing methods, the only way to demonstrate its advantage is through experimental results. However, based on the experiments, the proposed method is not significantly better than the existing methods. Sometimes it may be even worse than vanilla training without any techniques designed for label noise (e.g. Table 2).  Furthermore, some existing methods are missing in the experimental results, such as DivideMix, which I believe is a reasonable baseline without employing self-supervised learning.








**Summary Of The Paper:**

This paper proposes N-student learning to combat overfitting in neural network training such as that caused by label noise. The proposed method trains multiple neural models at the same time in a cross-fitting manner. Namely, each model is trained only on a subset of the training examples, where some of the training labels that are likely to be noisy are replaced with the predictions of other models. The authors experiment on multiple datasets to showcase their method.

**Summary Of The Review:**

As noted above, this paper lacks novelty and its quality and clarity also need to be improved. Significant revision is required to make it ready for publication.

---

### Decision · Program_Chairs · 2023-01-20

**Decision:**

Reject

**Justification For Why Not Higher Score:**

Limited novelty and lack of deeper experimental analyses.

**Justification For Why Not Lower Score:**

N/A

**Metareview: Summary, Strengths And Weaknesses:**

The paper proposed n student learning to combat overfitting. The proposed method learns multiple neural networks, each trains on a subset of the full training data, and generates pseudo labels for learning to reduce overfitting to noisy labels. All of the reviewers expressed concerns regarding the limited novelty, i.e., the use of the multiple neural networks and pseudo labels to reduce overfitting has been widely studied. While the experiment results are promising, additional insights, analyses and ablation studies will be required to shed lights on why and how the proposed method worked better than existing approaches.